# Learning to Adapt via Latent Domains for Adaptive Semantic Segmentation

**Yunan Liu, Shanshan Zhang**,* **Yang Li, Jian Yang**
PCA Lab, Key Lab of Intelligent Perception and Systems
for High-Dimensional Information of Ministry of Education,
Jiangsu Key Lab of Image and Video Understanding for Social Security,
School of Computer Science and Engineering,
Nanjing University of Science and Technology, Nanjing, China
{liuyunan, shanshan.zhang, yangli1995, csjyang}@njust.edu.cn

## Abstract

Domain adaptive semantic segmentation aims to transfer knowledge learned from labeled source domain to unlabeled target domain. To narrow down the domain gap and ease adaptation difficulty, some recent methods translate source images to target-like images (latent domains), which are used as supplement or substitute to the original source data. Nevertheless, these methods neglect to explicitly model the relationship of knowledge transferring across different domains. Alternatively, in this work we break through the standard "source-target" one pair adaptation framework and construct multiple adaptation pairs (e.g. "source-latent" and "latent-target"). The purpose is to use the meta-knowledge (how to adapt) learned from one pair as guidance to assist the adaptation of another pair under a meta-learning framework. Furthermore, we extend our method to a more practical setting of open compound domain adaptation (a.k.a multiple-target domain adaptation), where the target is a compound of multiple domains without domain labels. In this setting, we embed an additional pair of "latent-latent" to reduce the domain gap between the source and different latent domains, allowing the model to adapt well on multiple target domains simultaneously. When evaluated on standard benchmarks, our method is superior to the state-of-the-art methods in both the single target and multiple-target domain adaptation settings.

## 1 Introduction

Semantic segmentation is a popular task in computer vision, which assigns pixel-wise semantic labels for given images. It has been widely utilized to facilitate downstream applications such as video surveillance and autonomous driving. Recent progress on image semantic segmentation has been driven by deep neural networks trained on a large amount of labeled data, which are yet expensive to obtain. An alternative way is to generate synthetic images with pixel-level ground truth readily available in an effortless way [1, 2]. However, the model purely trained on synthetic datasets usually performs rather poor on real data. To address this issue, domain adaptation methods are used to reduce the domain shift and learn domain-invariant representations by minimizing distribution discrepancy between source and target domains [3, 4]. Following the advances of generative adversarial nets (GANs) [5], adversarial learning has been used to align representations of different domains in an adversarial manner [6, 7, 8, 9]. Recent studies introduce an additional intermediate domain, i.e. latent domain, to narrow down the huge domain gap between source and target domains [10, 11, 12, 13, 14, 15, 16, 17, 18, 19, 20, 21]. This is achieved by image-to-image

---

*The corresponding author is Shanshan Zhang.

35th Conference on Neural Information Processing Systems (NeurIPS 2021).

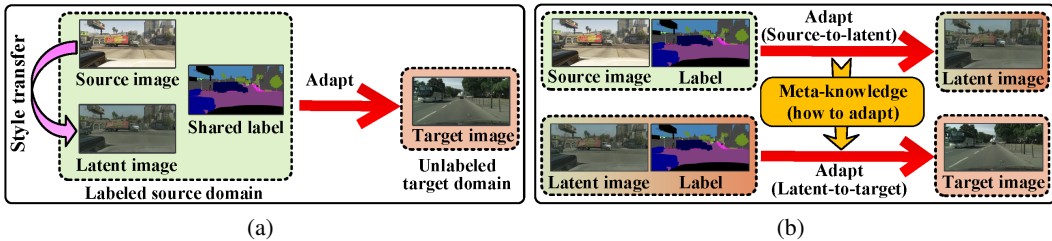

Figure 1: Comparison of different domain adaptation frameworks for semantic segmentation. (a) Previous frameworks built one domain adaptation pair that directly adapts a model from source domain and latent domain to target domain. In contrast, (b) we construct two domain adaptation pairs. The purpose is to use the meta-knowledge learned from one pair as guidance to assist the adaptation of another pair under a meta-learning framework.

translation technique, which generates augmented images that share the same contents as well as ground truth labels with the source domain and similar style to the target domain.

Previous works use the latent domain as supplement or substitute to the original source domain to ease the adaptation difficulty, following the traditional one-pair domain adaptation framework, as shown in Fig. 1(a). In contrast, in this paper we construct two domain adaptation pairs: "source→latent" and "latent→target". The purpose is to use the meta-knowledge learned from one pair of "source→latent" as guidance to assist the adaptation of another pair of "latent→target", as shown in Fig. 1(b). Specifically, we use a meta-learning framework to learn the meta-knowledge, which reveals that a model starting from which initial condition can adapt well on the adaptation pair of "source→latent". Meta-learning (a.k.a. learning to learn) has re-surged recently due to its efficacy for few-shot deep learning. In our paper, we only borrow the idea of bi-level optimization, to achieve an initialization that is good for both domain adaptation and segmentation. Please note for latent-target adaptation, since we do not have segmentation labels on target, we cannot optimize for the segmentation task. Thus, an initialization that is only good for domain adaptation is not sufficient; it is necessary to obtain an initialization that is good for both domain adaptation and segmentation. On the other hand, this initialization can be well transferred from source-latent to latent-target as it does not overfit to the domain adaptation or segmentation tasks on the latent domain thanks to the property of bi-level optimization. Recently, a couple of works have also successfully applied meta-learning to domain adaptation. The most related work to ours is [22], which requires multiple source domains or a proportion of labeled target data. In contrast, our proposed method makes use of a generated latent domain, and thus is more flexible in practice.

Most existing domain adaptation methods only aim at adapting a model to a single target domain, without considering a more practical scenario where the target consists of multiple data distributions. To investigate a more realistic setting for domain adaptation, we further study the problem of open compound domain adaptation (a.k.a multiple-target domain adaptation) [23]. In this setting, the target is a union of multiple domains without domain labels. To address this challenge, we extend our proposed method from single-target to multi-target domain adaptation. Specifically, we embed an additional domain pair of "latent→latent" to algin arbitrary two latent domains in the meta-training phase, which better learn the meta-knowledge of adaptation across different domains. In the meta-testing phase, the learned meta-parameters are used to initialize model parameters, promoting our model to adapt well on multiple target domains simultaneously.

In summary, the contribution of our work is three folds. (1) We propose a meta-learning framework compatible for both single-target and multi-target domain adaptation settings. To the best of our knowledge, this is the first work that adopts meta-learning framework to handle domain adaptive semantic segmentation. (2) We first generate additional images as a latent domain via style transfer from source to target domains. This latent domain is then used to construct domain pairs for meta-learning, which transfers meta-knowledge of adaptation across different domains. (3) Our approach achieves state-of-the-art performance on several challenging benchmark datasets. Also, we provide comprehensive model analyses for the proposed method.

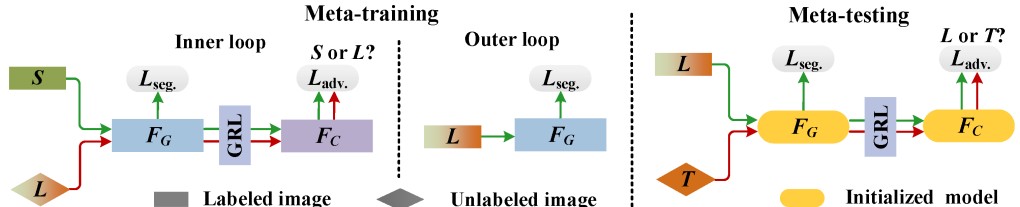

Figure 2: Training procedure of our meta-learning framework for single-target domain adaptation. In the meta-training phase, we construct a domain pair of "source→latent" to learn the meta-knowledge via bi-level optimization. The meta-knowledge reveals a model starting from which initial condition adapts well from labeled domain to a new unlabeled domain. In the meta-testing phase, we use the learned meta-knowledge as guidance to assist the adaptation of "latent→target".

## 2 Related Work

Since our method incorporates style transfer with meta-learning optimization for domain adaptive semantic segmentation, we will review related works from the above two aspects, respectively.

### 2.1 Domain Adaptive Semantic Segmentation Based on Style Transfer

Recent advances in domain adaptive semantic segmentation have highlighted the complementary role of pixel-level alignment, which is mainly achieved by image translation methods (e.g. style transfer) to translate images from one domain to another. Some works [10, 11, 12, 16, 17] construct a unified framework to learn image translation and domain adaptation in a sequential way. In contrast, BDL [13] proposes a bidirectional learning framework, consisting a closed loop to optimize the adaptation and image translation networks alternatively. Some other works [15, 18] investigate cross image translation between source and target domains, enforcing the model to produce consistent predictions for the original and translated images with the same contents. Instead of using adversarial learning, TGCF [14] constructs a self-ensembling learning framework, which is composed of a teacher network and a student network. To preserve the semantic information of translated image, FDA [19] and PCE [20] introduce Fourier transform to image translation. FDA [19] does not require any training to perform the domain alignment, just using a simple Fourier transform and its inverse. PCE [20] uses the phase consistency of Fourier transform to provide more flexibility to align the source domain and target domain. The DHA method [21] focuses on a more practical setting of multiple-target domain adaptation. It generates translated images to construct multiple latent domains, each of which contains the same "style" with a specific target domain. Then a couple of multiple domain classifiers are used to perform the target-to-source alignment separately. Existing methods ignore explicitly modeling the relationship of cross-domain knowledge transfer, which is expected to promote the adaptation performance.

### 2.2 Meta-Learning for Deep Neural Networks

Meta-learning for neural network has a long history, but have resurged in popularity recently. This has been largely driven by its efficacy for few-shot deep learning [24, 25], reinforcement learning [26], hyperparameter optimization [27], and neural architecture search [28]. More recently, meta-learning has also been successfully applied to domain adaptation [29, 30, 31, 22]. EAML [31] uses a meta-adaptation framework to learn representations for adapting the model to continually evolving target domains. Meta-online [22] introduces a new framework to enhance performance by learning the initial conditions (i.e. model parameter) of existing domain adaptation methods, whereby the proposed framework is only suitable for multi-source or semi-supervised domain adaption settings. We draw inspiration from MAML [32], and in particular from the idea of learning initial conditions [22] of neural network optimization that efficiently adapt the model to the target domain. The major difference to [22] is that we construct domain pairs using the generated latent domain images, which share the same contents with the source domain and a similar style with the target domain. Such an intermediate domain makes the meta-knowledge transfer become easier yet more effective.

# 3 Method

## 3.1 Preliminaries

We define the source domain as $D_S$ and the target domain as $D_T = \{D_T^0, D_T^1, ..., D_T^{K-1}\}$, where $K$ indicates the number of target domains. Assuming $K>1$, we take it as the multi-target domain setting; when $K=1$, it degenerates to a single target domain adaptation problem.

When assuming $K>1$, we first need to split the whole target data to multiple domains by discovering different styles. Here, we use the statistics of CNN features (i.e. mean and standard deviations) to represent the domain-specific style information following [21, 33]. Based on this representation, we then perform $k$-means clustering [34] to assign a domain label to each target image.

---

**Algorithm 1** Meta-Knowledge Learning for Single Target Domain Adaptation (STDA)

---

**Input:** Labeled data from $D_S$ and $D_L$, and unlabelled data from $D_T$.
**Initialise:** Model parameters $\Theta_0$, learning rate $\alpha$, $\beta$, $\gamma$, iteration number $I$ at each epoch, iteration
    number $J$ and $N$ in the meta-training and meta-testing phases, respectively.
  1: **for** $i = 1, 2, ..., I$ **do**
  2:     $\widetilde{\Theta}_0 = \Theta_{i-1}$;
  3:     Sample $(D_S)_i$, $(D_L)_i$ and $(D_T)_i$ uniformly from $D_S$, $D_L$ and $D_T$, respectively;
  4:     **for** $j = 1, 2, ..., J$ **do**
  5:         Sample $(D_S)_j$ and $(D_L)_j$ from $(D_S)_i$ and $(D_L)_i$, respectively;
  6:         $\widetilde{\Theta}_j = \widetilde{\Theta}_{j-1} - \alpha \bigtriangledown_{\widetilde{\Theta}_{j-1}} L_{uda}(\widetilde{\Theta}_{j-1}; (D_S)_j, (\overline{D_L})_j)$;     \\ Inner loop
  7:     **end for**
  8:     $\Theta_0 = \Theta_{i-1} - \beta \bigtriangledown_{\Theta_{i-1}} L_{seg}(\widetilde{\Theta}_J; (D_L)_i)$;     \\ Outer loop for learning initial condition
  9:     **for** $n = 1, 2, ..., N$ **do**
10:         Sample $(D_L)_n$ and $(D_T)_n$ from $(D_L)_i$ and $(D_T)_i$, respectively.
11:         $\Theta_n = \Theta_{n-1} - \gamma \bigtriangledown_{\Theta_{n-1}} L_{uda}(\Theta_{n-1}; (D_L)_n, (D_T)_n)$     \\ Meta-testing
12:     **end for**     $\Theta_i = \Theta_N$
13: **end for**
14: **return** $\Theta_I$

---

Bases on one source domain and $K(K \geq 1)$ target domains, we use an image translation model $G$ to generate augmented images, constructing $K$ additional intermediate domains, i.e. latent domains. For example, the generated image on the $k$-th latent domain denoted as $X_{L^k}^i$ shares the same content with the $i$-th source image $X_S^i$ and possesses similar style information with $z$-th image $X_{T^k}^z$ on the $k$-th target domain, and can be formulated as: $X_{L^k}^i = G(X_S^i, X_{T^k}^z)$, where $z$ is a random index, indicating randomly selecting an image from the target domain. Here, we use a recently proposed image translation method (i.e. PCE [20]) as $G$ to generate augmented images. In order to preserve the semantic information, PCE introduces the Fourier transform and designs a phase consistency loss for image translation. Please refer to [20] and [21] for more details.

For the basic pairwise domain adaptation module, we use the adversarial learning strategy. Specifically, the semantic segmentation network $F_G$ and domain classification network $F_C$ are optimized via the following unsupervised domain adaptation (UDA) loss $L_{uda}$:

$$L_{uda} = L_{seg} + \lambda L_{adv}, \tag{1}$$

where $L_{seg}$ is the cross-entropy loss using ground truth annotations on the source domain; $L_{adv}$ is the adversarial loss that optimizes $F_G$ and fools $F_C$ by maximizing the probability of misclassifying the source and target data; and $\lambda$ is the weight used to balance the two terms. For adversarial learning, we simply insert a gradient reversal layer (GRL) [40]) between $F_G$ and $F_C$, allowing the whole model to be optimized in an end-to-end manner.

In contrast to the traditional one-pair framework (as shown in Fig. 1(a)), in this work we propose a meta-learning framework that consists of multiple domain pairs (as shown in Fig. 1(b)), aiming at learning the meta-knowledge (how to adapt) as guidance for improving the adaption performance. In the following subsections 3.2 and 3.3, we will describe our meta-learning framework for single-target and multi-target domain adaptation in detail, respectively.

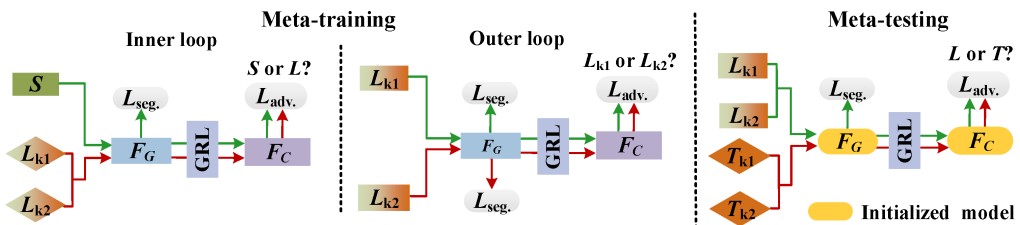

Figure 3: Training procedure of our meta-learning framework for multiple-target domain adaptation. We randomly select data from two latent domains $L_{k1}$ and $L_{k2}$ in the meta-training phase, and data from two target domains $T_{k1}$ and $T_{k2}$ in the meta-testing phase. In the outer loop, we construct an additional domain adaption pair of "latent→latent", and define the supervised domain adaptation as the optimization goal, allowing to learn the meta-knowledge that reveals a model starting from which initial condition generalizes on multiple unlabeled domains. In the meta-training phase, we use the learned meta-knowledge to initialize model parameters, promoting our model to adapt well on multiple target domains simultaneously.

### 3.2   Meta-Knowledge Learning for Single-Target Domain Adaptation

In this section, we describe the proposed meta-learning framework for single-target domain adaptation (STDA). As shown in Fig. 2, the proposed framework consists of two pairs of domain adaptation: "source→latent" and "latent→target". In the meta-training phase, we apply the meta-optimization on the pair of "source→latent" to learn the meta-knowledge (i.e. initial condition in our case), revealing that a base UDA model starting from which initial parameters can adapt well from labeled data (i.e. source domain) to unlabeled data (i.e. latent domain). In the meta-testing phase, the learned meta-knowledge are used to guide the optimization of "latent→target" domain adaptation.

(1) Meta-training phase. The optimization procedure of meta-learning for initial condition can be considered as a problem of bi-level optimization as follow:

$$\Theta = \overbrace{\underset{\theta}{\arg\min}\, L_{outer}(\underbrace{L_{inner}(\Theta; D_{support})}_{\text{Inner loop}}, D_{query})}^{\text{Outer loop}} \tag{2}$$

where $L_{inner}(\Theta, D_{support})$ and $L_{outer}(\Theta^*, D_{query})$ are the losses computed on support set $D_{support}$ and query set $D_{query}$, respectively, and the outer loop optimization starts from $\Theta^* = \arg\min L_{inner}$ once inner loop optimization has been completed. The goal of Eq. 2 is thus to learn the initial condition of base model such that it achieves minimum loss on both support set and query set. When both losses are differentiable, the meta-gradient in Eq. 2 involves a gradient through a gradient, which allows gradient update of outer loop to guide the inner loop optimization.

In the setting of unsupervised domain adaptation, the target data cannot be used as $D_{query}$ for outer loop optimization due to lack of annotations. To obtain some additional labeled data serving as $D_{query}$, we use an image translation model to generate augmented images (i.e. latent domain), which share the same contents and labels as the source images, as described in Section 3.1. We describe the training procedure of the proposed meta-learning for single-target domain adaption in Algorithm 1. In the inner loop, we replace $L_{inner}$ with the UDA loss in Eq. 1 as $L_{inner} := L_{uda}$, and assume there is no labels on the latent domain, making the model to adapt from the labeled source domain to the unlabelled latent domain. In the outer loop, we use the latent images with ground truth labels as the query set, which are used to evaluate the adaptation performance via a segmentation loss $L_{outer} := L_{seg}$. In this way, we aim to learn an initial condition $\Theta_0$ that enables our base domain adaptation model to adapt effectively on the "source→target" domain pair:

$$\Theta_0 = \overbrace{\underset{\theta_0}{\arg\min} \sum_{D_S, \overline{D_L}, D_L} L_{seg}(\underbrace{L_{uda}(\Theta_0; D_S, \overline{D_L})}_{\text{Inner loop}}, D_L)}^{\text{Outer loop}} \tag{3}$$

where $\overline{D_L}$ and $D_L$ are the identical data from latent domain, while the ground truth labels are not used for $\overline{D_L}$.

---

**Algorithm 2** Meta-Knowledge Learning for Multiple Target Domain Adaptation (MTDA)

---

**Input:** Labeled data $D_S$ from source domain, labeled data $D_L = \{D_L^0, D_L^1, ..., D_L^{K-1}\}$ from $K$ latent domains, and unlabeled data $D_T = \{D_T^0, D_T^1, ..., D_T^{K-1}\}$ from $K$ target domains.
**Initialise:** Model parameters $\Theta_0$, iteration number $I$, $J$ and $N$, learning rate $\alpha$, $\beta$, $\gamma$.

1: **for** $i = 1, 2, ..., I$ **do**
2:     $\widetilde{\Theta}_0 = \Theta_{i-1}$;
3:     **for** $j = 1, 2, ..., J$ **do**
4:         $\widetilde{\Theta}_j = \widetilde{\Theta}_{j-1} - \alpha \bigtriangledown_{\widetilde{\Theta}_{j-1}} L_{inner}(\widetilde{\Theta}_{j-1}; (D_S)_j, \overline{(D_L^{k1,k2})}_j)$;     \\ Inner loop
5:     **end for**
6:     $\Theta_0 = \Theta_{i-1} - \beta \bigtriangledown_{\Theta_{i-1}} (L_{seg}(\widetilde{\Theta}_J; (D_L^{k1,k2})_i) + L_{adv}(\widetilde{\Theta}_J; (D_L^{k1})_i, (D_L^{k2})_i))$; \\ Outer loop
7:     **for** $n = 1, 2, ..., N$ **do**
8:         $\Theta_n = \Theta_{n-1} - \gamma \bigtriangledown_{\Theta_{n-1}} L_{uda}(\Theta_{n-1}; (D_L^{k1,k2})_n, (D_T^{k1,k2})_n)$     \\ Meta-testing
9:     **end for**     $\Theta_i = \Theta_N$
10: **end for**
11: **return** $\Theta_I$

---

(2) Meta-testing phase. The parameters $\Theta_0$ learned from meta-training is used to initialize the domain adaptation model for "latent→target". The optimization can be formulated as follows:

$$\Theta = \arg\min_{\theta} L_{uda}(\Theta_0; D_L, D_T). \tag{4}$$

As shown in Algorithm 1, we obtain the model parameters $\Theta_N$ after meta-testing optimization at the $i$-th iteration; at the following $(i+1)$-th iteration, we use $\Theta_N$ to initialize the model such that $\widetilde{\Theta}_0 = \Theta_{i-1} = \Theta_N$ (Step 2 in Algorithm 1). When the entire training procedure finishes, we obtain an adapted semantic segmentation network $F_G$ (as shown in Fig. 2), which is expected to predict reasonable segmentation results on the target domain.

### 3.3 Meta-Knowledge Learning for Multiple-Target Domain Adaptation

We propose a meta-learning framework consisting of multiple domain adaptation pairs (i.e. "source→latent", "latent→latent", and "latent→target") for multiple-target domain adaptation (MTDA), promoting the adapted model to perform well on multiple target domains simultaneously. Fig. 3 shows the training procedure.

In the setting of MTDA ($K > 1$), we have labeled data on the source domain $D_S$ and generated latent domains $D_L = \{D_L^0, D_L^1, ..., D_L^{K-1}\}$, and attempt to adapt the UDA model to unlabeled target domain $D_T = \{D_T^0, D_T^1, ..., D_T^{K-1}\}$. In the meta-training phase, we define the optimization goal is to learn the meta-knowledge that a model starting from which initial condition can adapt well from a source domain to multiple latent domains. Expect for applying bi-level optimization on "source→latent", we also consider the domain adaptation pair of "latent→latent" to reduce domain shift between arbitrary two latent domains. The motivation of such a design can be represented by a toy example, as shown in Fig. 4. Let $A$, $B$, and $C$ indicate the center of data distribution on the source domain, and two latent domains, respectively. Suppose $A$, $B$, and $C$ locate at the three vertices of a triangle, our goal can be simplified to shorten the sides of $AB$ and $AC$ simultaneously. One simple strategy is to directly shorten the above sides, corresponding to applying domain adaptation on the pair of "source→latent". Another strategy (corresponding to domain adaptation on the pair of "latent→latent") is to shorten the side of $BC$, so that $AB$ and $AC$ can be further shortened indirectly. Since both strategies contribute to reduce $d(A, B)$ and $d(A, C)$ ($d$ for distance), we thus use two domain adaptation pairs consisting of "source→latent" and "latent→latent" in the meta-training phase to learn the meta-knowledge.

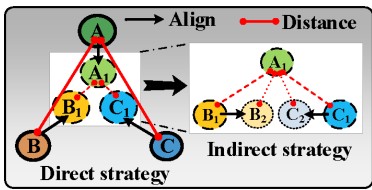

Figure 4: A toy example for reducing the length of AB and AC.

We describe the training procedure of the proposed meta-learning for multiple-target domain adaption in Algorithm 2, and introduce the optimization detail about the meta-training phase and the meta-testing phase as following:

(1) Meta-training phase. At each iteration, we randomly select samples $D_L^{k1,k2}=\{D_L^{k1}, D_L^{k2}\}$ from two latent domains, where $k1, k2 \in (0, 1, ..., K-1)$ and $k1 \neq k2$. In the inner loop, $F_C$ is used to discriminate the representation from $F_G$ belonging to source domain $D_S$ or latent domain $D_L^{k1,k2}$. The loss function $L_{inner}$ of the inner loop can be formulated as follows:

$$L_{inner}(\Theta_0; D_S, \overline{D_L^{k1,k2}}) = L_{seg}(\Theta_0; D_S) + L_{adv}(\Theta_0; D_S, \overline{D_L^{k1,k2}})), \qquad (5)$$

where $\overline{D_L^{k1,k2}}$ indicates latent data which is assumed without labels. In the outer loop, we let $F_C$ discriminate different latent domains. Suppose $\Theta^*$ denotes the initial condition of outer loop optimization as $\Theta^* = \arg\min L_{inner}$, the whole optimization goal of meta-training is defined as:

$$\Theta_0 = \arg\min_{\theta_0} \underbrace{\sum (L_{seg}(\Theta^*; D_L^{k1,k2}) + L_{adv}(\Theta^*; D_L^{k1}, D_L^{k2}))}_{L_{outer}(\Theta^*; D_L^{k1,k2})}, \qquad (6)$$

where $\Theta_0$ is the initial condition (i.e. model parameters) need to be optimized via meta-training optimization for the following meta-testing phase. As shown in Eq.6, the outer loop optimization can be be recognized as solving a problem of supervised domain adaptation between two latent domains, allowing the model to generalize on different latent domains. Specifically, the loss function of outer loop $L_{outer}$ in Eq.6 can be divided into two terms: the former term $L_{seg}$ takes the role of evaluating the segmentation performance on the query set (i.e. $D_L^{k1}$ and $D_L^{k2}$), and the latter one $L_{adv}$ enforces the model to learn domain-invariant representation for different latent domains by applying discrimination on the domain pair of ''latent$\rightarrow$latent''.

(2) Meta-testing phase. The initial condition $\Theta_0$ contains the meta-knowledge of from which initial condition that a UDA model can do well in transferring domain-specific knowledge from a labeled source domain to multiple unlabeled latent domains simultaneously. As shown in Fig. 3, we initialize the same UDA model with the learned $\Theta_0$ as the initial condition, and let $F_C$ discriminate latent from target domains. Then, the optimization goal of meta-testing on the domain pair of "latent$\rightarrow$target" is defined as follow:

$$\Theta = \arg\min_{\theta}(L_{uda}(\Theta_0; D_L^{k1,k2}, D_T^{k1,k2})) \qquad (7)$$

Where $D_L^{k1}$ and $D_L^{k2}$ share the similar style to $D_T^{k1}$ and $D_T^{k2}$, respectively. When the entire training procedure finishes, we obtain an adapted semantic segmentation network $F_G$ (as shown in Fig. 3) with parameters $\Theta$, which is expected to predict reasonable segmentation results on the multiple target domains simultaneously.

## 4 Experiments

### 4.1 Experimental Settings

**Data preparation.** In our experiments, we evaluate the proposed method on both single-target and multiple-target settings of unsupervised domain adaptation. Specifically, we take the GTA5 dataset [1] as the labeled source domain, while the Cityscapes [35] and C-Driving [21] datasets are adopted as unlabeled single-target domain and multi-target domains, respectively. For the multiple-target setting on the C-Driving dataset, we use the compound data containing "cloudy", "rainy", and "snowy" images as target domain for training. During testing, we evaluate the proposed method on the compound test set and a new unseen open test set containing "overcast" images. For image preprocessing, we resize the shorter side of images to 720 and randomly crop a patch from one image at each iteration with size 600×600. Besides, horizontal flip is applied as data augmentation. At each iteration, we randomly select 4 samples from each domain for training.

**Implementation.** We implement our proposed methods using the PyTorch v1.2.0 on a single NVIDIA P40 GPU (24G memory). For fair comparison, we adopt the DeepLabV2 [36] with VGG16 [37] as the segmentation network $F_G$. For the domain classification network $F_C$, we use an architecture similar to [6], which consists of 5 conv. layers with kernel 4×4 and stride of 2, and the numbers of channels are $\{64, 128, 256, 512, 1\}$. The hyper-parameter $\lambda$ in Eq. 1 is set to 0.01. The value of "J" and "N" (Algorithm 1 and 2) are set to 2 in our experiments. We use the SGD optimizer with 0.9 momentum and $5 \times 10^{-5}$ weight decay. The learning rates $\alpha$, $\beta$, $\gamma$ are empirically set to $1 \times 10^{-4}$, $5 \times 10^{-5}$, and $1 \times 10^{-4}$. For evaluation, images from target test set are resized to 1024×512 as input and the evaluation metric (i.e. IoU) is calculated on predictions upsampled to the original size.

Table 1: Comparison with state-of-the-art UDA methods (VGG16) on single-target domain adaptation (i.e. GTA5 to Cityscapes). The performance is measured by the intersection-over-union (IoU) for each class and mean IoU (mIoU). "Δ" denotes the improvement compared to source only.

| Method | road | sidewalk | building | wall | fence | pole | light | sign | veg. | terrain | sky | person | rider | car | truck | bus | train | motor. | bike | mIoU | Δ |
|---|---|---|---|---|---|---|---|---|---|---|---|---|---|---|---|---|---|---|---|---|---|
| Source only | 61.0 | 18.5 | 66.2 | 18.0 | 19.6 | 19.1 | 22.4 | 15.5 | 79.6 | 28.5 | 58.0 | 44.5 | 1.7 | 66.6 | 14.1 | 1.1 | 0.0 | 3.2 | 0.7 | 28.3 | -/- |
| CyCADA [11] | 85.2 | 37.2 | 76.5 | 21.8 | 15.0 | 23.8 | 22.9 | 21.5 | 80.5 | 31.3 | 60.7 | 50.5 | 9.0 | 76.9 | 17.1 | 28.2 | 4.5 | 9.8 | 0.0 | 35.4 | 7.1 |
| LSE [44] | 86.0 | 26.0 | 76.7 | 33.1 | 13.2 | 21.8 | 30.1 | 16.5 | 78.8 | 25.8 | 74.7 | 50.6 | 18.7 | 81.8 | 22.5 | 30.5 | 12.3 | 16.9 | 25.4 | 39.0 | 10.7 |
| CrCDA [42] | 86.8 | 37.5 | 80.4 | 30.7 | 18.1 | 26.8 | 25.3 | 15.1 | 81.5 | 30.9 | 72.1 | 52.8 | 19.0 | 82.1 | 25.4 | 29.2 | 10.1 | 15.8 | 3.7 | 39.1 | 10.8 |
| BDL [13] | 89.2 | 40.9 | 81.2 | 29.1 | 19.2 | 14.2 | 29.0 | 19.6 | 83.7 | 35.9 | 80.7 | 54.7 | 23.3 | 82.7 | 25.8 | 28.0 | 2.3 | 25.7 | 19.9 | 41.3 | 13.0 |
| PIT [38] | 86.2 | 35.0 | 82.1 | 31.1 | 22.1 | 23.2 | 29.4 | 28.5 | 79.3 | 31.8 | 81.9 | 52.1 | 23.2 | 80.4 | 29.5 | 26.9 | 30.7 | 20.5 | 1.2 | 41.8 | 13.5 |
| FDA [19] | 86.1 | 35.1 | 80.6 | 30.8 | 20.4 | 27.5 | 30.0 | 26.0 | 82.1 | 30.3 | 73.6 | 52.5 | 21.7 | 81.7 | 24.0 | 30.5 | 29.9 | 14.6 | 24.0 | 42.2 | 13.9 |
| LTIR [17] | 92.5 | 54.5 | 83.9 | 34.5 | 25.5 | 31.0 | 30.4 | 18.0 | 84.1 | 39.6 | 83.9 | 53.6 | 19.3 | 81.7 | 21.1 | 13.6 | 17.7 | 12.3 | 6.5 | 42.3 | 14.0 |
| DTST [41] | 88.1 | 35.8 | 83.1 | 25.8 | 23.9 | 29.2 | 28.8 | 28.6 | 83.0 | 36.7 | 82.3 | 53.7 | 22.8 | 82.3 | 26.4 | 38.6 | 0.0 | 19.6 | 17.1 | 42.4 | 14.1 |
| TGCF [14] | 90.2 | 51.5 | 81.1 | 15.0 | 10.7 | 37.5 | 35.2 | 28.9 | 84.1 | 32.7 | 75.9 | 62.7 | 19.9 | 82.6 | 22.9 | 28.3 | 0.0 | 23.0 | 25.4 | 42.5 | 14.2 |
| LDR [18] | 90.1 | 41.2 | 82.2 | 30.3 | 21.3 | 18.3 | 33.5 | 23.0 | 84.1 | 37.5 | 81.4 | 54.2 | 24.3 | 83.0 | 27.6 | 32.0 | 8.1 | 29.7 | 26.9 | 43.6 | 14.3 |
| FADA [43] | 92.3 | 51.1 | 83.7 | 33.1 | 29.1 | 28.5 | 28.0 | 21.0 | 82.6 | 32.6 | 85.3 | 55.2 | 28.8 | 83.5 | 24.4 | 37.4 | 0.0 | 21.1 | 15.2 | 43.8 | 14.5 |
| PCE [20] | 90.2 | 44.7 | 82.0 | 28.4 | 28.4 | 24.4 | 33.7 | 35.6 | 83.7 | 40.5 | 75.1 | 54.4 | 28.2 | 80.3 | 23.8 | 39.4 | 0.0 | 22.8 | 30.8 | 44.6 | 16.3 |
| **Our STDA** | **93.1** | **57.8** | **84.1** | **31.4** | **36.1** | **27.5** | **21.2** | **37.5** | **85.5** | **44.5** | **83.8** | **53.8** | **16.4** | **82.2** | **21.7** | **44.9** | **0.0** | **13.2** | **30.2** | **45.5** | **17.2** |

Table 2: Comparison with state-of-the-art methods (ResNet101) for single-target domain adaptation. The performance is measured by the intersection-over-union (IoU) for each class and mean IoU (mIoU). "Δ" denotes the improvement compared to source only.

| Method | road | sidewalk | building | wall | fence | pole | light | sign | veg. | terrain | sky | person | rider | car | truck | bus | train | motor. | bike | mIoU | Δ |
|---|---|---|---|---|---|---|---|---|---|---|---|---|---|---|---|---|---|---|---|---|---|
| Source only | 34.8 | 14.9 | 53.4 | 15.7 | 21.5 | 29.7 | 35.5 | 18.4 | 81.9 | 13.1 | 70.4 | 62.0 | 34.4 | 62.7 | 21.6 | 10.7 | 0.7 | 34.9 | 35.7 | 34.3 | -/- |
| LSE [44] | 90.2 | 40.0 | 83.5 | 31.9 | 26.4 | 32.6 | 38.7 | 37.5 | 81.0 | 34.2 | 84.6 | 61.6 | 33.4 | 82.5 | 32.8 | 45.9 | 6.7 | 29.1 | 30.6 | 47.5 | 13.2 |
| PLCA [47] | 84.0 | 30.4 | 82.4 | 35.3 | 24.8 | 32.2 | 36.8 | 24.5 | 85.5 | 37.2 | 78.6 | 66.9 | 32.8 | 85.5 | 40.4 | 48.0 | 8.8 | 29.8 | 41.8 | 47.7 | 13.4 |
| BDL [13] | 91.0 | 44.7 | 84.2 | 34.6 | 27.6 | 30.2 | 36.0 | 36.0 | 85.0 | 43.6 | 83.0 | 58.6 | 31.6 | 83.3 | 35.3 | 49.7 | 3.3 | 28.8 | 35.6 | 48.5 | 14.2 |
| CrCDA [42] | 92.4 | 55.3 | 82.3 | 31.2 | 29.1 | 32.5 | 33.2 | 35.6 | 83.5 | 34.8 | 84.2 | 59.0 | 32.2 | 84.7 | 40.6 | 46.1 | 2.1 | 31.1 | 32.7 | 48.6 | 14.3 |
| DTST [41] | 90.6 | 44.7 | 84.8 | 34.3 | 28.7 | 31.6 | 35.0 | 37.6 | 84.7 | 43.3 | 85.3 | 57.0 | 31.5 | 83.8 | 42.6 | 48.5 | 1.9 | 30.4 | 39.0 | 49.2 | 14.9 |
| LDR [18] | 90.8 | 41.4 | 84.7 | 35.1 | 27.5 | 31.2 | 38.0 | 32.8 | 85.6 | 42.1 | 84.9 | 59.6 | 34.4 | 85.0 | 42.8 | 52.7 | 3.4 | 30.9 | 38.1 | 49.5 | 15.2 |
| CCM [48] | 93.5 | 57.6 | 84.6 | 39.3 | 24.1 | 25.2 | 35.0 | 17.3 | 85.0 | 40.6 | 86.5 | 58.7 | 28.7 | 85.8 | 49.0 | 56.4 | 5.4 | 31.9 | 43.2 | 49.9 | 15.6 |
| FADA [43] | 91.0 | 50.6 | 86.0 | 43.4 | 29.8 | 36.8 | 43.4 | 25.0 | 86.8 | 38.3 | 87.4 | 64.0 | 38.0 | 85.2 | 31.6 | 46.1 | 6.5 | 25.4 | 37.1 | 50.1 | 15.8 |
| LTIR [17] | 92.0 | 55.0 | 85.3 | 34.2 | 31.1 | 34.9 | 40.7 | 34.0 | 85.2 | 40.1 | 87.1 | 61.0 | 31.1 | 82.5 | 32.3 | 42.9 | 0.3 | 36.4 | 46.1 | 50.2 | 15.9 |
| CAG [49] | 90.4 | 51.6 | 83.8 | 34.2 | 27.8 | 38.4 | 25.3 | 48.4 | 85.4 | 38.2 | 78.1 | 58.6 | 34.6 | 84.7 | 21.9 | 42.7 | 41.1 | 29.3 | 37.2 | 50.2 | 15.9 |
| FDA [19] | 92.5 | 53.3 | 82.4 | 26.5 | 27.6 | 36.4 | 40.6 | 38.9 | 82.3 | 39.8 | 78.0 | 62.6 | 34.4 | 84.9 | 34.1 | 53.1 | 16.9 | 27.7 | 46.4 | 50.5 | 16.2 |
| PCE [20] | 91.0 | 49.2 | 85.6 | 37.2 | 29.7 | 33.7 | 38.1 | 39.2 | 85.4 | 35.4 | 85.1 | 61.1 | 32.8 | 84.1 | 45.6 | 46.9 | 0.0 | 34.2 | 44.5 | 50.5 | 16.2 |
| PIT [38] | 87.5 | 43.4 | 78.8 | 31.2 | 30.2 | 36.3 | 39.9 | 42.0 | 79.2 | 37.1 | 79.3 | 65.4 | 37.5 | 83.2 | 46.0 | 45.6 | 25.7 | 23.5 | 49.9 | 50.6 | 16.3 |
| **Our STDA** | **88.4** | **50.8** | **82.7** | **39.4** | **24.9** | **34.6** | **43.7** | **46.6** | **84.3** | **38.6** | **81.7** | **61.3** | **41.9** | **77.8** | **50.4** | **39.0** | **5.4** | **40.3** | **53.2** | **51.9** | **17.6** |

## 4.2 Comparison with State of the Art

We evaluate our method on two unsupervised domain adaptation settings: single target and multi-target. The results are presented in Tables 1, 2 and 3, respectively.

For the single target setting, in Table 1 we compare our method with 12 state-of-the-art methods. All the methods are based on VGG16 [37] backbone. It can be seen that our method for single-target domain adaptation (STDA) achieves 45.5% on mIoU, and obtains 17.2% improvement compared to the source only model, both of which perform favorably against previous state-of-the-art methods. In Table 2, we compare our method with 13 state-of-the-art methods, which all use ResNet101 [39] as the backbone. It can be seen that our method achieves 51.9% on mIoU, outperforming previous state-of-the-art methods. These results further demonstrate that our proposed method achieves consistent top results on different backbones.

For the multi-target setting, we compare our method with two recently proposed multi-target methods [21, 23] and several state-of-the-art single-target methods [4, 6, 45, 46], following [21, 23]. In Table 3 we see that the proposed method for multi-target domain adaptation (MTDA) outperforms previous methods by a large margin w.r.t mIoU (4%), establishing a new state of the art. Besides, we retrain the PCE [20] model under the setting of MTDA, which achieves 31.1% w.r.t mIoU on the C-Driving dataset. These results indicate that our method outperforms PCE on both STDA and MTDA settings.

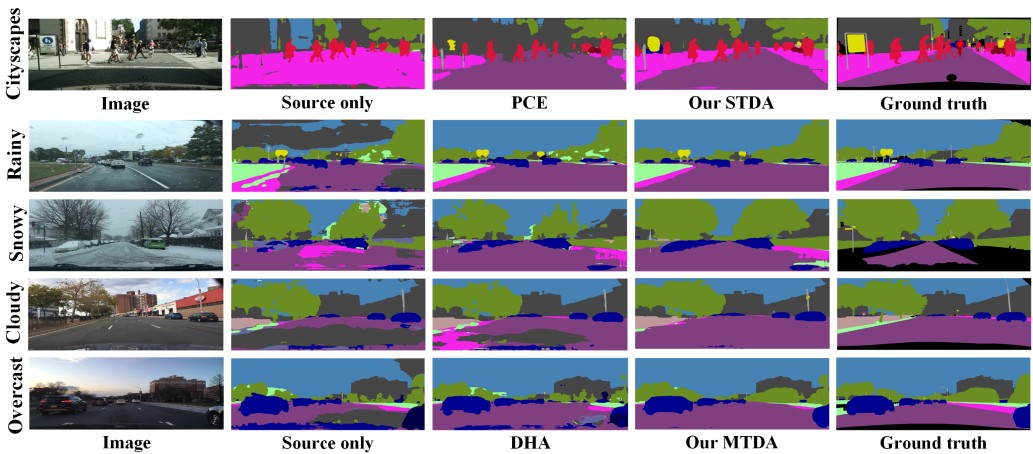

| | Image | Source only | PCE | Our STDA | Ground truth |

Figure 5: Qualitative results of ours and previous top methods. For single-target domain adaptation, we compare our STDA with PCE [20] on Cityscapes (1st row). For multi-target domain adaptation, we compare our MTDA with DHA [21] on C-Driving (from 2nd to 5th rows are "rainy", "snowy", "cloudy" and "overcast", respectively).

Table 3: Comparison with state-of-the-art U-DA methods on multi-target domain adaptation (i.e.GTA5 to C-Driving). The performance is measured by the intersection-over-union (IoU) for eachclass and mean IoU (mIoU). "Δ" denotes the improvement compared to source only.

| Method | Compound (C) | | | Open (O) | Average | |
| | rainy | snowy | cloudy | overcast | C+O | Δ |
| --- | --- | --- | --- | --- | --- | --- |
| Source only | 16.2 | 18.0 | 20.9 | 21.2 | 19.1 | -/- |
| AdaptSeg [6] | 20.2 | 21.2 | 23.8 | 25.1 | 22.5 | 3.4 |
| CBST [45] | 21.3 | 20.6 | 23.9 | 24.7 | 22.6 | 3.5 |
| IBN-Net [46] | 20.6 | 21.9 | 26.1 | 25.5 | 23.5 | 4.4 |
| PyCDA [4] | 21.7 | 22.3 | 25.9 | 25.4 | 23.8 | 4.7 |
| OCDA [23] | 22.0 | 22.9 | 27.0 | 27.9 | 25.0 | 5.9 |
| DHA [21] | 27.0 | 26.3 | 30.7 | 32.8 | 29.2 | 10.1 |
| Our source only | 19.7 | 20.7 | 22.4 | 24.3 | 22.5 | -/- |
| **Our MTDA** | **31.5** | **30.2** | **33.0** | **35.0** | **33.2** | **10.7** |

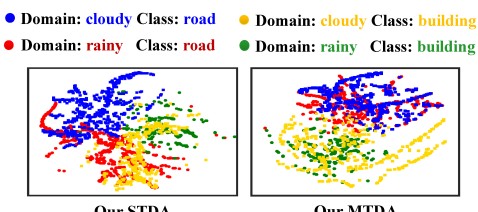

● Domain: cloudy Class: road  ● Domain: cloudy Class: building
● Domain: rainy  Class: road  ● Domain: rainy  Class: building

Our STDA          Our MTDA

Figure 6: The t-SNE visualization of feature space. For each category (e.g. road/building), the features are better aligned by our MTDA than STDA.

We also provide some qualitative semantic segmentation results in Fig. 5, where we observe obvious improvements against both source only and previous top methods. These quantitative and qualitative results demonstrate the effectiveness of our method for unsupervised domain adaptation.

## 4.3 Model Analysis

To verify the effectiveness of our methods, we provide comparisons on different ways of using the latent domain, as shown in Table 4.

From the results, we have the following observations: (1) *The latent domain eases adaptation via narrowing down the domain gap.* Compared to ⓐ, other methods, using the latent domain in different ways, all perform better. (2) *The latent domain can be used to construct multiple pairs, which performs better than the standard one pair.* Compared to ⓒ, ⓓ and ⓔ obtain significant improvements ($\sim$ 3%) by constructing two DA pairs, which distributes the adaptation difficulty on two procedures. (3) *Meta-knowledge transition between DA pairs helps more than simple finetuning.* In ⓕ, our proposed STDA framework learns meta-knowledge of how to adapt from labeled data to unlabeled data, which is more effective to assist the adaptation of "latent→target" than the finetuning strategy used in ⓔ. (4) *When the target data distributes sparsely, our multi-target method further improves the performance.* Our MTDA improves over STDA on the C-Driving dataset by modeling the target data as multiple domains.

Table 4: Model analysis on framework design. "$A{\to}B$" indicates adapting a model from $A$ to $B$. ⓐⓑⓒ: standard one pair UDA using Eq. 1; ⓓ: similar to ⓒ but using two independent domain classifiers for $S{\to}T$ and $L{\to}T$; ⓔ: first trained on $S{\to}T$ and then finetuned on $L{\to}T$ at each iteration; ⓕ: our single-target domain adaptation (STDA) method (Fig.2), which uses the meta-knowledge learned from $S{\to}L$ to assist the adaptation of $L{\to}T$; ⓖ: our multi-target domain adaptation (MTDA) method (Fig.3).

| Method GTA5→ | Number of DA pairs | Single-target (→Cityscapes) | Multi-target (→C-Driving) | | | | |
|---|---|---|---|---|---|---|---|
| | | | rainy | cloudy | snowy | overcast | Average |
| ⓐ. $S{\to}T$ | 1 | 35.7 | 24.3 | 21.9 | 26.4 | 27.6 | 25.8 |
| ⓑ. $L{\to}T$ | 1 | 37.2 | 24.7 | 23.6 | 28.3 | 29.6 | 27.5 |
| ⓒ. $(S, L){\to}T$ | 1 | 38.1 | 25.0 | 24.8 | 29.0 | 30.1 | 28.2 |
| ⓓ. $S{\to}T$ and $L{\to}T$ | 2 | 40.8 | 27.9 | 26.4 | 29.5 | 31.6 | 29.7 |
| ⓔ. $(S{\to}L){\to}(L{\to}T)$ | 2 | 41.9 | 29.4 | 27.7 | 29.5 | 31.9 | 30.4 |
| ⓕ. Our STDA (Fig.2) | 2 | 45.5 | 31.3 | 30.2 | 32.6 | 34.4 | 32.6 |
| ⓖ. Our MTDA (Fig.3) | 3 | -/- | 31.5 | 30.2 | 33.0 | 35.0 | 33.2 |

We further analyze the feature space learned with our MTDA and STDA. In Fig. 6, We provide the t-SNE visualization [50] of extracted features from $F_G$ using our STDA and MTDA, respectively. It appears that our MTDA yields more generalized domain-invariant features. More specifically, the feature distributions of different domains are better aligned in MTDA than that in STDA.

## 5 Conclusion

In this paper, we propose a novel meta-learning framework for single-target and multi-target domain adaptive semantic segmentation. In particular, we generate latent data via an image translation model and construct multiple domain adaptation pairs. The purpose is to use the meta-knowledge learned from some pairs as guidance to assist the adaptation of "latent-to-target" under a meta-learning framework. Experimental results show the effectiveness of our method, which establishes new state-of-the-art performance on both settings of single-target and multi-target domain adaptation.

## 6 Acknowledgments

This work was supported by the National Natural Science Foundation of China (Grant No. 62172225), the Funds for International Cooperation and Exchange of the National Natural Science Foundation of China (Grant No. 61861136011), the Fundamental Research Funds for the Central Universities (Grant No.30920032201), and the National Key Research and Development Program of China under Grant 2017YFC0820601.

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
