# [Supplementary Material]
# Learning to Adapt via Latent Domains for Adaptive Semantic Segmentation

## A    Appendix

In the supplementary material, we provide more experimental results summarized as follows:

- In A.1, we use ResNet101 as the backbone network and compare our method with state-of-the-art methods, demonstrating that our method achieves consistent top results on different backbones.

- In A.2, we provide more t-SNE visualization results for a comprehensive analysis on the feature space learned from different models.

- In A.3, we study the effect of the image-to-image translation model on the performance of domain adaptive semantic segmentation.

- In A.4, we discuss the limitations of our method and provide the URL link of code to reproduce the main experimental results.

### A.1    Comparison with State-of-the-art Methods

In the main paper, we report results using VGG16 as the backbone for both settings: single-target and multi-target domain adaptation. Here we further provide comparisons on ResNet101 [1].

For the single-target setting, in Table 2, we compare our method with 13 state-of-the-art methods, which all use ResNet101 as the backbone. It can be seen that our method for single-target domain adaptation (STDA) achieves 51.9% on mIoU, outperforming previous state-of-the-art methods. These results further demonstrate that our proposed method achieves consistent top results on different backbones.

Table 1: Comparison with state-of-the-art methods on multi-target domain adaptation. "$V$" and "$R$" indicate the method using VGG16 and ResNet101 backbone networks, respectively.

| Method | Backbone | rainy | snowy | cloudy | overcast | Average |
|--------|----------|-------|-------|--------|----------|---------|
| OCDA [2] | $V$ | 22.0 | 22.9 | 27.0 | 27.9 | 25.0 |
| DHA [3] | $V$ | 27.0 | 26.3 | 30.7 | 32.8 | 29.2 |
| **Our MTDA** | $V$ | **31.5** | **30.2** | **33.0** | **35.0** | **33.2** |
| **Our MTDA** | $R$ | **32.3** | **33.3** | **39.2** | **41.9** | **37.9** |

For the multi-target setting, since previous methods do not provide results on ResNet101, in Table 1 we report our results on ResNet101 to show that further improvements of 4.7% (w.r.t mIoU) can be obtained on a stronger backbone.

We also provide some qualitative semantic segmentation results in Figure. 1, where we observe obvious improvements against the source only method. These quantitative and qualitative results demonstrate the effectiveness of our method built on different backbone networks.

Submitted to 35th Conference on Neural Information Processing Systems (NeurIPS 2021). Do not distribute.

Table 2: Comparison with state-of-the-art methods (ResNet101) for single-target domain adaptation.

| Method | road | sidewalk | building | wall | fence | pole | light | sign | veg. | terrain | sky | person | rider | car | truck | bus | train | motor. | bike | mIoU |
|---|---|---|---|---|---|---|---|---|---|---|---|---|---|---|---|---|---|---|---|---|
| LSE [4] | 90.2 | 40.0 | 83.5 | 31.9 | 26.4 | 32.6 | 38.7 | 37.5 | 81.0 | 34.2 | 84.6 | 61.6 | 33.4 | 82.5 | 32.8 | 45.9 | 6.7 | 29.1 | 30.6 | 47.5 |
| PLCA [5] | 84.0 | 30.4 | 82.4 | 35.3 | 24.8 | 32.2 | 36.8 | 24.5 | 85.5 | 37.2 | 78.6 | 66.9 | 32.8 | 85.5 | 40.4 | 48.0 | 8.8 | 29.8 | 41.8 | 47.7 |
| BDL [6] | 91.0 | 44.7 | 84.2 | 34.6 | 27.6 | 30.2 | 36.0 | 36.0 | 85.0 | 43.6 | 83.0 | 58.6 | 31.6 | 83.3 | 35.3 | 49.7 | 3.3 | 28.8 | 35.6 | 48.5 |
| CrCDA [7] | 92.4 | 55.3 | 82.3 | 31.2 | 29.1 | 32.5 | 33.2 | 35.6 | 83.5 | 34.8 | 84.2 | 58.9 | 32.2 | 84.7 | 40.6 | 46.1 | 2.1 | 31.1 | 32.7 | 48.6 |
| DTST [8] | 90.6 | 44.7 | 84.8 | 34.3 | 28.7 | 31.6 | 35.0 | 37.6 | 84.7 | 43.3 | 85.3 | 57.0 | 31.5 | 83.8 | 42.6 | 48.5 | 1.9 | 30.4 | 39.0 | 49.2 |
| LDR [9] | 90.8 | 41.4 | 84.7 | 35.1 | 27.5 | 31.2 | 38.0 | 32.8 | 85.6 | 42.1 | 84.9 | 59.6 | 34.4 | 85.0 | 42.8 | 52.7 | 3.4 | 30.9 | 38.1 | 49.5 |
| CCM [10] | 93.5 | 57.6 | 84.6 | 39.3 | 24.1 | 25.2 | 35.0 | 17.3 | 85.0 | 40.6 | 86.5 | 58.7 | 28.7 | 85.8 | 49.0 | 56.4 | 5.4 | 31.9 | 43.2 | 49.9 |
| FADA [11] | 91.0 | 50.6 | 86.0 | 43.4 | 29.8 | 36.8 | 43.4 | 25.0 | 86.8 | 38.3 | 87.4 | 64.0 | 38.0 | 85.2 | 31.6 | 46.1 | 6.5 | 25.4 | 37.1 | 50.1 |
| LTIR [12] | 92.0 | 55.0 | 85.3 | 34.2 | 31.1 | 34.9 | 40.7 | 34.0 | 85.2 | 40.1 | 87.1 | 61.0 | 31.1 | 82.5 | 32.3 | 42.9 | 0.3 | 36.4 | 46.1 | 50.2 |
| CAG [13] | 90.4 | 51.6 | 83.8 | 34.2 | 27.8 | 38.4 | 25.3 | 48.4 | 85.4 | 38.2 | 78.1 | 58.6 | 34.6 | 84.7 | 21.9 | 42.7 | 41.1 | 29.3 | 37.2 | 50.2 |
| FDA [14] | 92.5 | 53.3 | 82.4 | 26.5 | 27.6 | 36.4 | 40.6 | 38.9 | 82.3 | 39.8 | 78.0 | 62.6 | 34.4 | 84.9 | 34.1 | 53.1 | 16.9 | 27.7 | 46.4 | 50.5 |
| PCE [15] | 91.0 | 49.2 | 85.6 | 37.2 | 29.7 | 33.7 | 38.1 | 39.2 | 85.4 | 35.4 | 85.1 | 61.1 | 32.8 | 84.1 | 45.6 | 46.9 | 0.0 | 34.2 | 44.5 | 50.5 |
| PIT [16] | 87.5 | 43.4 | 78.8 | 31.2 | 30.2 | 36.3 | 39.9 | 42.0 | 79.2 | 37.1 | 79.3 | 65.4 | 37.5 | 83.2 | 46.0 | 45.6 | 25.7 | 23.5 | 49.9 | 50.6 |
| **Our STDA** | 88.4 | 50.8 | 82.7 | 39.4 | 24.9 | 34.6 | 43.7 | 46.6 | 84.3 | 38.6 | 81.7 | 61.3 | 41.9 | 77.8 | 50.4 | 39.0 | 5.4 | 40.3 | 53.2 | 51.9 |

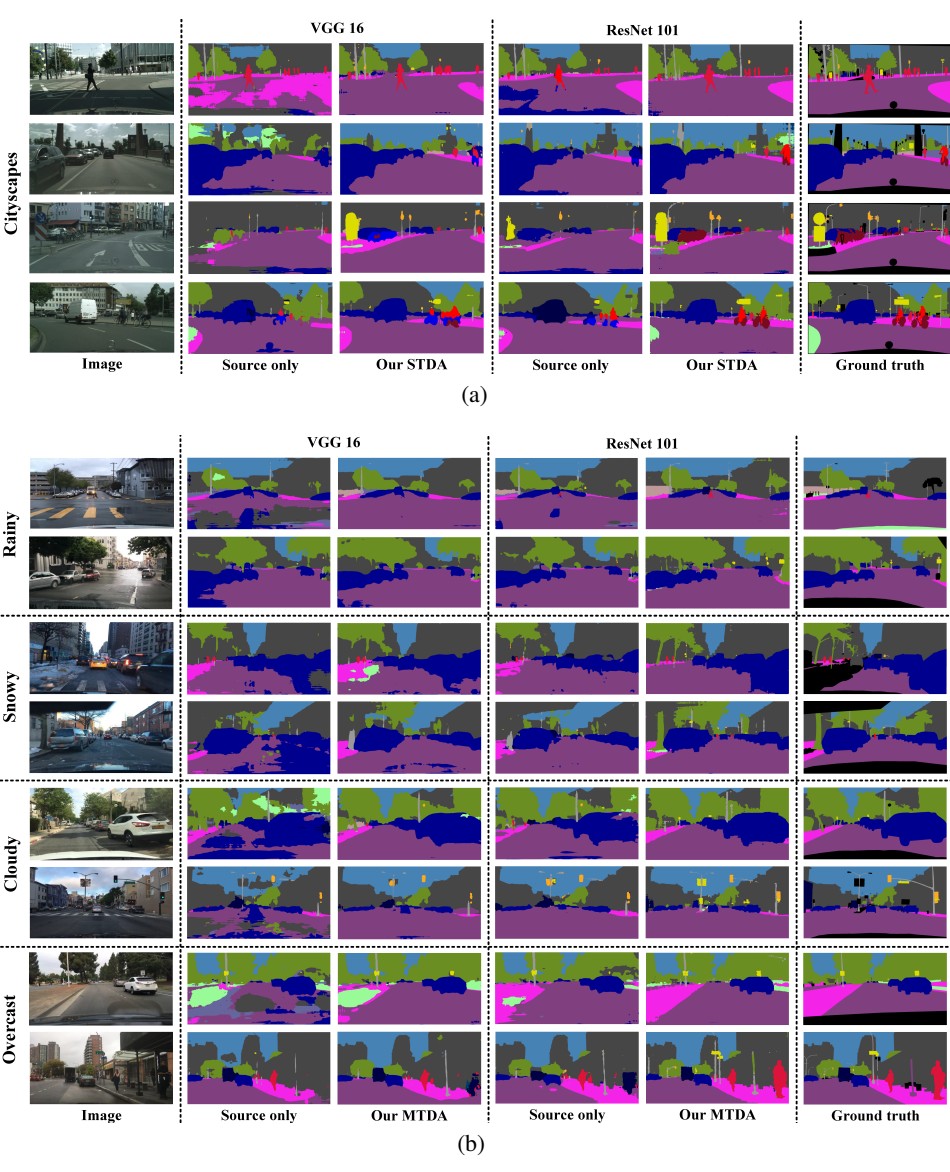

Figure 1: Qualitative results of source only model and our method using VGG16 and ResNet101 backbones. (a) Single-target domain adaptation (i.e. GTA5→Cityscapes), and (b)multi-target domain adaptation (i.e. GTA5→C-Driving).

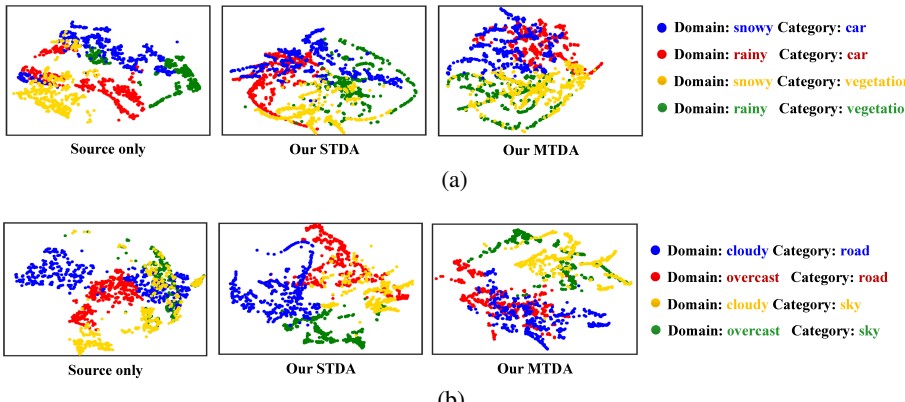

(a)

(b)

Figure 2: Comparison of feature space from source only, our STDA and MTDA models. In (a) and (b), the representations of different categories on different domains are mapped to 2-D space via the t-SNE toolbox.

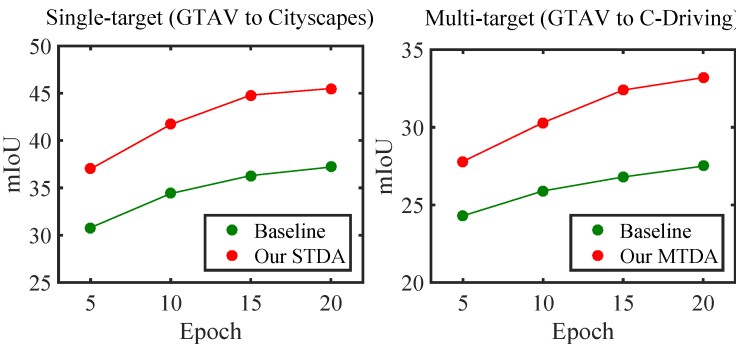

Figure 3: Effect of the image translation quality on domain adaptive semantic segmentation. The x-axis indicates the training stages of image translation model, and the y-axis is the performance of domain adaptation using the latent data from different image translation models.

## A.2 Contrastive Analysis on t-SNE Visualization Results

Following [3, 17], we use the t-SNE [18] to map the high-dimensional features learned from different models to a 2-D space shown in Figure 2. From the results, we have the following observations: (1) *The feature distributions of different categories are better distinguished in our STDA and MTDA than that in source only model.* Compared to the source only model, our STDA and MTDA learn more discriminative representations for different categories, yielding more accurate predictions for semantic segmentation. (2) *The feature distributions of different domains are better aligned in MTDA than that in STDA.* Compared to STDA, our MTDA enforces the alignment of domain-level representations during outer-level optimization, yields more generalized domain-invariant features.

## A.3 Study the Effect of Image Translation on Domain Adaptation

We conduct an experiment to study the effect of image-to-image translation model on domain adaptation. For different image translation methods, the performance of image translation is hard to evaluate by a standard metric. However, for a specific image translation method (e.g. the PCE model [15] used in our method), the translation performance is assumed to improve over training. In Figure 3, we provide the performance of domain adaptation at different training phase of PCE, where the baseline method directly performs domain adaptation without meta-learning on the pair of "latent→target".

From the results in Figure 3, we have two observations: (1) With the training of image translation continues, the performance of image translation becomes better, promoting the performance of

different domain adaptation methods. (2) The performance gain of our method compared to baseline model is consistent, demonstrating the robustness of our method that utilizes augmented latent images from different image translation models.

## A.4 Discussion

The limitations of our proposed method lie in the following two aspects: (1) Although the inference time of our method is similar to previous works when using the same backbone network, our method costs more time to train the meta-learning framework (roughly takes 8 hours for each epoch during the STDA training). (2) As discussed in Section A.3, different image translation methods affects the performance of domain adaptation, and thus we rely on a strong image translation method to achieve good performance.

To reproduce our main experimental results, we release the code at: Code link. The experimental environments are listed as follows:

- Ubuntu 16.04 environment (Python 3.6, CUDA10.0, CuDNN7)
- PyTorch=1.2.0 installed following the official instructions (https://pytorch.org)
- Dependencies: pip install opencv-python/tqdm/yacs>=0.1.5