# OpenReview forum: "Learning to Adapt via Latent Domains for Adaptive Semantic Segmentation"
_NeurIPS.cc/2021/Conference — NeurIPS 2021 Poster_

### Official Review · Reviewer_y8x3 · 2021-07-12

**Rating:** 7
**Confidence:** 5

**Summary:**

The paper works on the domain adaptation task for semantic segmentation with either one target domain or multiple target domains. In order to solve this problem, the author proposed a novel meta-learning framework to explore the better initialization for the adaptive model and achieve a clear improvement compared to other SOTA methods.

**Limitations And Societal Impact:**

Yes

**Main Review:**

Overall speaking, the paper is good. It is easy to read and the proposed method is novel. The experiments were well designed and further complemented by the supplementary material. My main concerns about the submission are listed as follows.

1. I wonder whether the author used any self-training methods in the paper to achieve the results shown in Table 1 of the main paper and Table 2 in the supplementary material. I think most of the baselines in both tables are achieved via self-training kind of techniques by generating pseudo-labels in various ways. If the author does not use self-training, I will be astonished by the performance and hope to see what the results will be if self-training is further added. If the author used self-training, I would like to see the comparison to the baselines without using any self-training technique  to further evaluate the achivement given by the proposed method.

2. The mechasim of meta-learning used for domain adaptation is not very clear to me. In classicial meta-learning,  the meta-training often contains seveval datasets and let the network learn how to adapt to novel dataset. Thus during meta-testing phase, the model can fit to the test dataset in just a few iterations. However, in the paper the meta-training only contains a single dataset pair for the single target domain adaptataion case. So I think the model is not fully trained such that it knows how to fit to a novel dataset. This problem can be solved by training the model with multiple dataset pairs which is missing in the paper. I hope the author can further show enough explanation and it will be great if the author can provide more experimental results for the case I mentioned above.

3. The adaptation performance for ResNet-101 is more ciritcal and the author might need to consider to move the results in supplementary material to the main paper.


final comments after rebuttal

After carefully reading the paper and discussing it with the authors, my concern is well addressed. Although other reviewers question the novelty of this paper about using meta-learning, meta-learning is easier to be used for domain generalization or multi-source domain adaptation problems, where there are multiple domains. I think domain adaptation is a different case and generating a latent domain to help build the meta-learning setting is still novel. Thus I raise my rating and suggest accepting this paper.

**Time Spent Reviewing:**

2 hours

---

> ### Author Response · Authors · 2021-08-09
> **The responses to the Reviewer y8x3**
>
> The responses to the Reviewer y8x3
>
> **Comment 1**: About the self-training strategy in the paper.
>
> **Response 1**: Generally, previous methods for domain adaptive semantic segmentation can be roughly classified into two groups: adversarial learning based methods and self-training based methods. Among the compared methods in Table 1 of the main paper, six of them adopt adversarial learning and another six use self-training strategy that generates pseudo-labels for unlabeled target images.
> In this paper, our base model for domain adaptation only adopts adversarial learning (as presented in Eq.(1)), and does not require to generate pseudo-labels for unlabeled target images.  That is to say, we do not use self-training. According to your comments, we conduct an additional experiment to evaluate the performance of our model when the self-training strategy is further used. Specifically, we use the source only segmentation network to generate pseudo-labels for target images, and compute the segmentation loss on the target data at the meta-testing phase. After adding self-training strategy, our STDA using VGG16 and ResNet 101 as backbone network further improves by around 0.4%, achieving 45.87% and 52.41% w.r.t mIoU on the Cityscapes dataset, respectively. We will add these experimental results in the final version.
>
> **Comment 2**:  The mechasim of meta-learning used for domain adaptation is not very clear to me.
>
> **Response 2**: For meta-learning methods (e.g. Meta-online [22], and MAML [32], etc.) that learn the initial condition, the basic unit of model optimization is “task” rather than “dataset”. In our proposed meta-learning framework, the optimization process consists of three stages, each of which com putes gradients on a specific task. Specifically, at each iteration, (1) the inner loop computes DA (domain adaptation) loss for the task of adaptation from labeled source data to latent data; (2) the outer loop computes segmentation loss for the segmentation task on the latent data; (3) the meta-testing uses the model parameters learned from the meta-training to initiate the model, and computes the DA loss on the task of adaptation from labeled latent data to unlabeled target data. Taking the STDA as an example, the goal of our meta optimization is to make the model learn the knowledge from one task (DA of source-latent), and then use the learned knowledge to assist a new task (DA of latent-target) but not to simply transfer knowledge across different datasets.
>
> **Comment 3**:  The author might need to consider to move the results in supplementary material to the main paper.
>
> **Response 3**: Thank you for your kind comments, we will move the results in supplementary material to the main paper in the final version.

---

> > ### Comment · Reviewer_y8x3 · 2021-08-14
> > **following question**
> >
> > After reading the rebuttal, I wonder whether the author could further clarify the working mechanism of the meta-learning technique proposed in the paper. Specifically, in algorithm 1, what I can see is the model parameters are first updated with L6 given by the source and latent samples and then update with L8 based on the segmentation loss. I think this part is the initialization part. Then the model is further learned with the latent and target samples in the manner shown in L11. The whole process is iterated across all the training samples with J=N=2. I don't quite understand why the parameters learned with L6 and L8 are a good initialization for the downstream task. I think in L6 and L8, the model is trained to decrease the domain discrepancy between source and latent. But I don't see how it clearly promote the domain alignment for latent and target domain.

---

> > > ### Author Response · Authors · 2021-08-15
> > > **The responses to the following question from Reviewer y8x3**
> > >
> > > Following the classical MAML framework [32], we use a bi-level optimization (from Line4 to Line8) to learn the meta knowledge from source-latent adaptation. In Line 8,  $\Theta_0$ consists the knowledge learned from both domain adaptation (optimized in Line 6) and segmentation (optimized in Line 8) tasks, and thus can serve as an initialization for adaptive segmentation on other domains. Specifically, we use the gradient computed from the segmentation loss (as shown in Line8) to update the initial parameter  $\Theta_{i-1}$ from Line2 rather than $\Theta_{J-1}$, involving a gradient through a gradient. On the other hand, $\Theta_0$ consists general knowledge as the domain adaptation and the segmentation tasks are optimized in a sequential manner. Compared with simultaneous optimization of two tasks, this bi-level optimization is helpful to improve the generalization ability of learned knowledge (thus called meta knowledge), avoiding overfitting on the current domain. Please note this is the key idea of MAML framework [32]. Therefore,  $\Theta_0$ updated in Line 8 is a good initialization for the latent-target adaptation. The general knowledge learned from source-latent adaptation can be well transferred to the latent-target adaptation.
> > >
> > > Please do not hesitate to contact us if you have any further questions.

---

> > > > ### Comment · Reviewer_y8x3 · 2021-08-15
> > > > **following question**
> > > >
> > > > Thanks for your response. However, I still question the meta-learning setting proposed in this paper. First, In [32], the author clearly said the meta-learning is designed for the few-shot problem, which is not the case for domain adaptation. Second, the hypothesis for meta-learning is the different tasks form a distribution (P(T) in [32]) that can cover the target task. But in the submission, as I mentioned previously, the meta-training process only contains one domain pair- source->latent, which I view as a single task and I hardly think it can describe any task distribution. Could you further clarify?

---

> > > > > ### Author Response · Authors · 2021-08-16
> > > > > **The responses to the reviewer y8x3**
> > > > >
> > > > > Yes, in [32] MAML is specifically designed for few-show learning. However, we need to note the key component of MAML -bi-level optimization- is to obtain a general gradient direction, that is compatible for multiple different tasks. The task distribution p(T) is more specific for few-shot learning, aiming to let the model learn general knowledge of recognition.
> > > > > Here, in our paper, we only borrow the idea of bi-level optimization, to achieve an initialization that is good for both domain adaptation and segmentation. Please note for latent-target adaptation, since we do not have segmentation labels on target, we cannot optimize for the segmentation task. Thus, an initialization that is only good for domain adaptation is not sufficient; it is necessary to obtain an initialization that is good for both domain adaptation and segmentation. On the other hand, this initialization can be well transferred from source-latent to latent-target as it does not overfit to the domain adaptation or segmentation tasks on the latent domain thanks to the property of bi-level optimization.

---

### Official Review · Reviewer_fSWY · 2021-07-12

**Rating:** 4
**Confidence:** 4

**Summary:**

This paper proposes a meta-learning-based UDA framework for semantic segmentation, which adopts meta-knowledge learned from source-latent adaptation to teach latent-target adaptation gradually.

**Limitations And Societal Impact:**

Yes, they have addressed the limitation and potential negative societal impact.

**Main Review:**

This paper is well organized and the method is explained in details. But there are stills some limitations in this work:
 1. The idea is not novel enough. On the one hand, meta-learning strategy has been applied to UDA for classification. On the other hand, meta-knowledge learning looks very similar to pre-training and fine-tuning strategy, both of whom aim to alleviate the difficulty of direct transfer from source to target. The only difference is that this method adopts an iterative strategy to update the model.

2. The method is somewhat complicated, which incorporates transfer learning and adversarial learning. This will increase the complexity of model and impedes reproducibility. In addition, the author build many adaptation pairs, such as"source to latent", "latent to target", and "latent to latent"pairs, which makes the model too difficult to train.

 3. The experimental results are not convincing. The authors claim that this paper achieves SOTA on several benchmarks, but some works are missed in this paper, e.g., Self-supervised Augmentation Consistency for Adapting Semantic Segmentation (CVPR 2021), which achieves better performance. In addition, the authors only use VGG as backbone, while recent works adopt resnet101. I think the authors would better compare with these new methods with the same setting, such as
 (1) Prototypical Pseudo Label Denoising and Target Structure Learning for Domain Adaptive Semantic Segmentation (CVPR 2021).
 (2)Category anchor-guided unsupervised domain adaptation for semantic segmentation (NeurIPS2019)

 4. Since this model is suitable for open compound adaptation, it would be better to compare this method to other open compound adaptation methods, such as references [21]. [23].


**Time Spent Reviewing:**

2.5 hours

---

> ### Author Response · Authors · 2021-08-09
> **The responses to the Reviewer fSWY**
>
> The responses to the Reviewer fSWY
>
> **Comment 1**: On the one hand, meta-learning strategy has been applied to UDA for classification. On the other hand, meta-knowledge learning looks very similar to pre-training and fine-tuning strategy,
>
> **Response 1**: (1) Although meta-learning has been applied to domain adaptive image classification, previous works require either multiple labeled source domains or a proportion of labeled target data, while in our work we transfer the meta knowledge regarding how to adapt via constructed latent domains, which is thus more applicable. Also, our method can be easily extended from single target to multi-target setting via building multiple domain pairs, as shown in our paper. (2) The meta-learning strategy in our work is significantly different from the fine-tuning strategy. The fine-tuning learning firstly optimizes the model on one task, and then continues to optimize it on another task, making the model optimization at two stages independent. In contrast, our proposed meta-learning framework consists of meta-training and meta-training phases, in which the meta-testing phase uses fine-tuning strategy while the meta-training phase applies a bi-level optimization on two tasks. As presented at Line 8 of Algorithm 1 and Line 6 of Algorithm 2, the gradients computed on the task (the segmentation on latent data) of outer loop will be used to optimize the model parameters on the task (the adaptation from source to latent) of inner loop. In this way, our meta-training can learn the meta knowledge, revealing that a model starting from which initial parameters can adapt well from labeled source data to unlabeled latent data. And then, the meta knowledge will assist the task (the adaptation from latent to target) in the meta-testing phase. To verify the advantage of our meta-learning over the fine-tuning strategy, we compare our meta-knowledge learning (Method “f”) with fine-tuning learning strategy (Method “e”) in Table 3, where we find that our meta knowledge learning (Method “f”) is more effective to assist the adaptation of “latent!target” than the fine-tuning strategy used in method “e”.
>
> **Comment 2**: The method is somewhat complicated, which incorporates transfer learning and adversarial learning. This will increase the complexity of model and impedes reproducibility. In addition, the author build many adaptation pairs, such as"source to latent", "latent to target", and "latent to latent"pairs, which makes the model too difficult to train.
>
> **Response 2**: Meta-learning has been widely and successfully used in many applications due to their powerful learning ability. In this work, we implement our meta-learning framework following the classical MAML framework [32]. The model optimization of our framework consists of three stages, each of which compute gradient on a specific task (a.k.a domain pair). Thanks to the excellent reproducibility of MAML, the model optimization through three stages in our framework becomes easy. In addition, we make the training easier by using the GRL layer to avoid multiple steps of training required by the conventional adversarial learning. Moreover, please note the meta-learning strategy does not introduces extra inference time at test time. Based on the above explanations, we argue that our proposed framework is in fact easy to reproduce as agreed by Review owPT .
>
> **Comment 3**: The authors claim that this paper achieves SOTA on several benchmarks, but some works are missed in this paper, e.g., [R1] CVPR 2021. In addition, the authors only use VGG as backbone, while recent works adopt resnet101. I think the authors would better compare with these new methods with the same setting, such as (1) [R2] CVPR 2021. (2) [R3] NeurIPS2019.
>
> **Response 3**: When we submit our paper to NeurIPS2021, the CVPR2021 papers have not been published, so we did not include them for comparison. In addition, please note that we have compared our method with other methods (including [R3]) using ResNet101 as backbone network, as shown in Table 1 and Table 2 in the supplementary material.
>
> **Comment 4**: Since this model is suitable for open compound adaptation, it would be better to compare this method to other open compound adaptation methods, such as references [21]. [23].
>
> **Response 4**: In fact, we have compared our method with those open compound adaptation methods including [21] and [23] in Table 2 of the main paper.

---

### Official Review · Reviewer_owPT · 2021-07-15

**Rating:** 6
**Confidence:** 4

**Summary:**

This work introduces a meta-learning framework, based on MAML, for single-target DA (STDA) and open compound DA (OCDA) in semantic segmentation. Similar to Li et al. [22], the idea is to learn the initial condition (model's parameters) that is more favorable for target adaptation. The proposed framework makes use of an existing image-2-image model to translate source images into, what are called here, latent images; this is to have target-like annotated data to guide meta-optimization.

Each iteration of the proposed algorithm is composed of two consecutive steps: (1) meta-training operating on source and latent data to find the favorable initial condition for the next step and (2) meta-testing to perform target adaptation starting from the learned initial condition.

The meta-training optimization, similar to MAML, involves inner and outer loops. In case of STDA, the inner loop performs UDA optimization steps on labeled source and unlabeled latent samples; the outer loop, based on the optimized weights of the inner loops, meta-optimizes the network to minimize segmentation loss computed using labeled latent data. In case of OCDA with k target sub-domains, k-mean clustering is first done to assign sub-domain pseudo-labels to target images. This information is used in the image-2-image operation to translate source data into sub-domain-like data. Different to STDA, there is an additional adversarial objective in the outer loop.

To evaluate performance, this work consider two DA benchmarks: GTA5->Cityscapes (STDA) and Cityscapes->C-Driving (OCDA). Reported results show improvement of the proposed framework as compared to previous approaches.

**Limitations And Societal Impact:**

The authors discussed the limitations of their work. Potential negative societal impact was not provided.

**Main Review:**

This is the first work adopting meta-learning for domain adaptation semantic segmentation. Improvement over state-of-the-art methods demonstrates the merit of the proposed framework. The paper presentation is quite clear and easy to follow. In general this is a decent work and I'm leaning toward the positive side. There are though still a few questions and concerns that I hope to receive satisfactory answers.

- What are the exact values of I,J and N (Algos 1 and 2) used in the experiments?

- In the OCDA experiment, clustering is done on features coming from which model? Using domain labels, may the authors provide the clustering performance? How does bad clustering performance affect the final results?

- The authors took the segmentation and discriminator models in [6] but instead of using the proposed GAN-like adversarial training in [6], the authors opted to GRL. What is the reason of this choice?

**Time Spent Reviewing:**

5 hours

---

> ### Author Response · Authors · 2021-08-09
> **The responses to the Reviewer owPT**
>
> The responses to the Reviewer owPT
>
> **Comment 1**: What are the exact values of I,J and N (Algos 1 and 2) used in the experiments?
>
> **Response 1**: The value of “I” is 6242, which is equal to the quotient of the number of training samples divided by batchsize (i.e. 4). Considering the limited computational memory, the value of “J” and “N” are set to 2 in our experiments. We will add the details of these settings in the final version.
>
> **Comment 2**:  About the clustering.
>
> **Response 2**: In this paper, we exactly follow the operation in OCDA [21] that uses the statistics of CNN features and k-means clustering to discover different styles in target domain. (1) Following [21], we use ImageNet pretrained VGG16 model to encode style of target images. Specifically, we use relu1_2 features. (2) In the supplementary material of OCDA [21], the authors have evaluated the clustering performance using the silhouette score as metric. (3) The clustering performance will affect the performance of domain adaptation. We have explored the upper bound performance when directly using domain labels, whereby our MTDA can achieve 33.92% w.r.t mIoU on the C-Driving dataset. We will add above details in the final version.
>
> **Comment 3**: What is the reason of using GRL?
>
> **Response 3**: The advantage of using GRL is that the entire network can be trained in an end-to-end manner.

---

### Official Review · Reviewer_Pvnj · 2021-07-16

**Rating:** 5
**Confidence:** 4

**Summary:**

The paper proposes a meta-learning framework for open compound domain adaptation. The method uses k-mean clustering to estimate the domain of target samples and employs the image translation method to augment datasets. Then the augmented images are used for the meta-learning framework. The method outperforms baselines on Cityscapes and C-Driving.

**Limitations And Societal Impact:**

Limitations are well addressed.

**Main Review:**


1. While the paper improves performance, the novelty is somewhat limited since the paper combines the existing image translation method for DA and meta-learning framework.
2. The paper is hard to read and explanations are not compact. Especially, notation is not very clear to understand.
3. The paper should explore the effect of k-means clustering. If there is no k-mean clustering (i.e., all target images are treated as single target domain), how well does the method work? At the same time, it would be better to provide accuracy of SOTA single target DA baselines (e.g., [20]) in Table 2.
4. No sensitivity analysis, no visualization of generated images, and no theoretical analysis.
5. How many runs are averaged in the experiments? (standard deviations?)
6. GTA5 is the only source domain in the experiments. Why don't use the Synthia dataset? In addition why there is a big gap between source-only models?


-- POST REBUTTAL --

I appreciate the effort for the response, but my concern about the experiment still remains. The experiments are performed only on one dataset as a source domain. I encourage authors to explore other datasets to verify the effectiveness of the method. In addition, other backbones (e.g., ResNet) should be explored. I also agree with the reviewer fSWY about concerns on the novelty. Therefore, I keep my score as weak reject.


**Time Spent Reviewing:**

4

---

> ### Author Response · Authors · 2021-08-09
> **The responses to the Reviewer Pvnj**
>
> The responses to the Reviewer **Pvnj**
>
> **Comment 1**:  About the novelty of the proposed method.
>
> **Response 1**: To the best of our knowledge, this is the first work that adopts meta-learning framework to handle domain adaptive semantic segmentation. Although meta-learning has been applied to domain adaptive image classification, previous works require either multiple labeled source domains or a proportion of labeled target data, while in our work we transfer the meta knowledge regarding how to adapt via constructed latent domains, which is thus more applicable for unsupervised domain adaptation.
>
> We construct latent domains using image translation methods. Although some previous works also used translated images for domain adaptation, our method differs from them in the following way: previous frameworks directly adapts a model learned from source domain and latent domain to target domain via one domain adaptation pair; in contrast, we construct two domain adaptation pairs such that the meta-knowledge learned from one pair can be used as guidance to assist the adaptation of another pair under a meta-learning framework. In this way, our method is more adaptive and thus achieves better domain adaptation results across different datasets.
>
> **Comment 2**: The paper is hard to read and explanations are not compact. Especially, notation is not very clear to understand.
>
> **Response 2**: Thank you for your kind comments. We will carefully check our submission, making the explanation of our paper easier to understand in the final version.
>
> **Comment 3**: About the effect of k-means clustering.
>
> **Response 3**: In this paper, we exactly follow the clustering operation in OCDA [21] that uses the statistics of CNN features and k-means method to discover different styles in target domain. If we get rid of k-mean clustering for multiple target domains, the task degenerates to single target domain adaptation (STDA). In Table 3 of the main paper, we have provided the results of our STDA (Algorithm 1) on the setting of multiple target DA. Specifically, our STDA (Method “f”) achieves 32.6% w.r.t mIoU on C-Driving dataset. Besides, we retrain the PCE model under the setting of STDA, which achieves 31.1% w.r.t mIoU on the C-Driving dataset. These results indicate that our method outperforms PCE on both STDA and MTDA settings. Following your suggestion, we will add these results in Table 2 of the final version.
>
> **Comment 4**: No sensitivity analysis, no visualization of generated images, and no theoretical analysis.
>
> **Response 4**: (1) Sensitivity analysis. In Section 4.3 of the main paper and Section A.3 of the supplementary material, we explore the effect of two sensitive settings (i.e. the usage of latent domains and the quality of generated images) on the performance of domain adaptation. (2) Visualization of generated images. As presented from Line116 to Line 119 of the main paper, our method uses the image translation method proposed in PCE [20] to generate latent images. Since image translation method is not the contribution of our method and the space is limited, we did not provide visualization of generated images. In the final version, we will provide some visualization results of image translation. (3) Theoretical analysis. The effectiveness of meta-learning has been proved by many previous works through theoretical analysis. In this paper, we focus on how to effectively integrate latent domains with meta-learning for domain adaptive semantic segmentation. Therefore, we describe the proposed framework in detail and demonstrate the effectiveness of our method through experiments
>
> **Comment 5**:  How many runs are averaged in the experiments? standard deviations?
>
> **Response 5**:  We trained our STDA and MTDA for single target and multiple target domain adaptation for three times, respectively. In the main paper, we provide the best results from three STDA and MTDA models, following previous works [19,20,21,41] in this field. The average/standard deviation of three STDA and MTDA models are 45.17/0.43 and 33.05/0.13, respectively. We will add these results in the final version.
>
> **Comment 6**: GTA5 is the only source domain in the experiments. Why don't use the Synthia dataset? In addition why there is a big gap between source-only models?
>
> **Response 6**: (1) All previous methods use GTA5 as labeled source domain for both single target and multi-target settings, while only a couple of single target domain adaptation methods use Synthia dataset as the source dataset. The reason is that GTA5 contains 19 semantic categories, consistent with the target datasets (i.e. Cityscapes and C-Driving). For a fair comparison, we only use GTA5 as the source dataset in the main paper. We will add the results of single target domain adaptation using Synthia dataset in the final version. (2) Although the source-only models are trained on the same source dataset, we usually observe differences in performance across different papers (e.g. 19.1% w.r.t mIoU in [23] and 26.4% w.r.t mIoU in [21]), due to the length of training time probably. In fact, we do not use any extra tricks for training the source only model, and have released the source-only model via the link provided in supplementary material (https://github.com/Anony123lyn/Domain-adaptive-semantic-segmentation).

---

> ### Author Response · Authors · 2021-09-02
> **The responses to the Reviewer Pvnj**
>
> **The responses to the Reviewer Pvnj**:
>
> **(1) About the backbones (e.g., ResNet)**. Please note that we have compared our method with other methods using ResNet101 as backbone network, as shown in Table 1 and Table 2 in the supplementary material.
>
> **(2) About the source dataset**. In fact, all MTDA methods [21,23] only use one dataset (i.e. GTA5) as the source domain. Besides, only the GTA5 dataset contains 19 categories, consistent with the target datasets (i.e. Cityscapes and C-Driving). For a fair and quick comparison, we only use GTA5 as the source dataset for STDA and MTDA simultaneously in the main paper.
>
> **(3) About the novelty**. In [32] MAML is specifically designed for few-show learning. However, we need to note the key component of MAML -bi-level optimization- is to obtain a general gradient direction, that is compatible for multiple different tasks. Here, in our paper, we only borrow the idea of bi-level optimization, to achieve an initialization that is good for both domain adaptation and segmentation. Please note for latent-target adaptation, since we do not have segmentation labels on target, we cannot optimize for the segmentation task. Thus, an initialization that is only good for domain adaptation is not sufficient; it is necessary to obtain an initialization that is good for both domain adaptation and segmentation. On the other hand, this initialization can be well transferred from source-latent to latent-target as it does not overfit to the domain adaptation or segmentation tasks on the latent domain thanks to the property of bi-level optimization. After the rolling discussion, the reviewer y8x3 has approved of the novelty of our work.

---

> ### Comment · Reviewer_Pvnj · 2021-09-02
> **Additional Feedback**
>
> Thanks for the response again.
> 1) I expect comparisons with baselines using ResNet on the open compound DA setting as well. In addition, Table 1 in the main paper shows very marginal improvements on single-target DA.
> 2) For single-target DA, many papers use Synthia for an additional source domain. Since the paper does not provide theoretical insights on how the proposed method helps domain adaptation, I believe extensive experiments are necessary to show the effectiveness.
> 3) Overall, I do not think that the novelty of the proposed method outweighs these flaws.

---

> > ### Author Response · Authors · 2021-09-03
> > **The responses to the Reviewer Pvnj**
> >
> > Thank you for your valuable suggestions!
> >
> > (1) The source only model using ResNet achieves 28.4% w.r.t mIoU on the open compound DA setting. Our MTDA outperforms the source-only model by 9.5% w.r.t mIoU.
> >
> > (2) When we apply our STDA framework on another UDA scenario of SYNTHIA to Cityscapes, we also outperform previous methods under different backbones. The detailed results are as follows:
> >
> > ResNet101 backbone: [Ours 55.8% / PCE 53.6% / LDR 53.1% / LTIR 49.3%]. VGG16 backbone: [Ours 50.1% / PCE 48.7% / LDR 41.1% / LTIR 43.8%].
> >
> > Thank you for your suggestions, we will add the above results in the final version.

---

### Decision · Program_Chairs · 2021-09-27

**Decision:**

Accept (Poster)

**Comment:**

After the discussion period this paper received mixed scores with one recommendation for rejection, one for acceptance, one leaning reject and one leaning accept. There was disagreement about the novelty of the proposed approach. The key novelty claimed by the authors is using a translated source image as a domain unto itself and performing meta learning to learn an adaptable model from source $\rightarrow$ latent such that further adaptation from latent $\rightarrow$ target should be more effective. All reviewers agreed that the image translation portion (i.e., generation of latent domain) was based on prior work. The debate then centered around whether the introduction of a meta learning objective using this translated image was sufficient contribution as well as whether that idea proved effective empirically. After considering all reviewer comments, discussion, author rebuttal and examining the paper, it seems that although prior work has used meta learning for domain generalization the specific use in this paper for the latent domain together with the ablation study in Table 3 showing the value of using the meta objective instead of just adapting directly from L$\rightarrow$T implies that this design decision is useful and novel. It does however seem that the study into the open compound target together with using existing clustering from [21] is less of a contribution and I would encourage the authors to revise the text to clarify their key contribution. Further, as promised please move the ResNet experiments to the main paper.